# Cognitive Bias and Diagnostic Errors among Physicians in Japan: A Self-Reflection Survey

**DOI:** 10.3390/ijerph19084645

**Published:** 2022-04-12

**Authors:** Takashi Watari, Yasuharu Tokuda, Yu Amano, Kazumichi Onigata, Hideyuki Kanda

**Affiliations:** 1General Medicine Center, Shimane University Hospital, Izumo 693-8501, Japan; 2Division of Hospital Medicine, University of Michigan Health System, Ann Arbor, MI 48105, USA; 3Okinawa Muribushi Project for Teaching Hospitals, Okinawa 901-2132, Japan; yasuharu.tokuda@gmail.com; 4Faculty of Medicine, Shimane University, Izumo 693-8501, Japan; amanoshimane@gmail.com (Y.A.); konigata@med.shimane-u.ac.jp (K.O.); 5Department of Public Health, Faculty of Medicine, Okayama University, Oakayama 700-8558, Japan; hkanda@okayama-u.ac.jp

**Keywords:** cognitive bias, diagnostic errors, physicians, self-reflection, survey

## Abstract

This cross-sectional study aimed to clarify how cognitive biases and situational factors related to diagnostic errors among physicians. A self-reflection questionnaire survey on physicians’ most memorable diagnostic error cases was conducted at seven conferences: one each in Okayama, Hiroshima, Matsue, Izumo City, and Osaka, and two in Tokyo. Among the 147 recruited participants, 130 completed and returned the questionnaires. We recruited primary care physicians working in various specialty areas and settings (e.g., clinics and hospitals). Results indicated that the emergency department was the most common setting (47.7%), and the highest frequency of errors occurred during night-time work. An average of 3.08 cognitive biases was attributed to each error. The participants reported anchoring bias (60.0%), premature closure (58.5%), availability bias (46.2%), and hassle bias (33.1%), with the first three being most frequent. Further, multivariate logistic regression analysis for cognitive bias showed that emergency room care can easily induce cognitive bias (adjusted odds ratio 3.96, 95% CI 1.16−13.6, *p*-value = 0.028). Although limited to a certain extent by its sample collection, due to the sensitive nature of information regarding physicians’ diagnostic errors, this study nonetheless shows correlations with environmental factors (emergency room care situations) that induce cognitive biases which, in turn, cause diagnostic errors.

## 1. Introduction

Diagnostic errors are among the most important issues related to patient safety and have been the subject of much research attention worldwide [1,2]. In this context, a diagnostic error is defined as “the failure to (a) establish an accurate and timely explanation of the patients’ health problem(s) or (b) communicate that explanation to the patient” [3]. In simple terms, it refers to diagnoses that are delayed, wrong, or altogether missed [1,3]. Diagnostic errors occur in at least 5% of outpatient department and emergency room cases, and lead to fatalities in approximately one in every 1000 cases, resulting in an estimated 40,000−120,000 deaths per year in the United States [3,4]. Estimates also indicate that economic losses caused by unnecessary testing, treatment, and deaths due to diagnostic errors amount to approximately 30% of the annual national healthcare expenditures in the United States [5]. While it is difficult to fully clarify the cognitive diagnostic processes used by physicians, such investigations have significantly improved over the last 20 years in conjunction with the field of cognitive psychology [6,7,8,9]. Based on ideas such as the dual-process model and early vs. late diagnosis approach, cognitive bias appears to be the most crucial factor in identifying the causes of diagnostic errors and proposing the next stage of diagnostic excellence by physicians [10,11,12,13,14,15]. Generally, a cognitive bias is defined as an intuitive heuristic leading to an adverse outcome. Although various cognitive biases are already known, evaluating which types are the root causes of diagnostic errors is important [14,15,16,17,18,19,20]. Despite the importance of examining the association between diagnostic errors and cognitive bias, as in the previous studies mentioned above, limited research has explored diagnostic errors and cognitive biases among physicians in Japan [21]. Therefore, we used diagnostic error cases, about which physicians can most vividly reflect with respect to the situation, environment, psychological background, and patient outcome, to clarify the characteristics of situational factors and cognitive biases associated with diagnostic errors in the Japanese medical setting.

## 2. Methods

### 2.1. Participants and Survey

A 19-item survey questionnaire was developed to obtain information on the experiences of diagnostic errors among Japanese physicians. We specifically recruited primary care physicians who worked in various specialty areas and settings (e.g., clinics and hospitals). To facilitate participant comprehension, we also provided them with a document explaining how to define diagnostic errors, classify cognitive biases, and analyze the situational factors described later. Participants filled out the survey themselves; the survey was used to collect information on cases defined as the most vividly recalled cases of clear diagnostic errors in a physician’s career, while obtaining detailed situational and environmental information related to the same. We set aside 3 min to recall the settings, and 20 min to fill in the details. The anonymity of participants and the institutions was guaranteed for all participants who provided their informed consent before participation.

Regarding sample size, we assumed that cognitive bias was involved in 50% of diagnostic errors that occurred in the emergency room, a challenging environment, and that 25% of diagnostic errors occurred elsewhere. For a desired power of 0.80 and alpha of 0.05, we estimated that we would need 124 participants to detect the differences. A total of 155 participants was considered as the target sample size, excluding 15% of the participants who did not agree to engage in the study and 5% of the participants who answered inappropriately. The first author held a series of workshops with the same content from July 2018 to January 2019, in which participants could reflect on their diagnostic error cases on the basis of clinical reasoning and thus learn from the errors. This paper-based data collection process was conducted immediately before the workshop at the place or hospital where the participants worked. Participant recruitment was conducted at conferences in seven different areas of Japan (i.e., Okayama University Hospital in Okayama, Hiroshima City Hospital in Hiroshima, Matsue Seikyo Hospital in Matsue, Shimane University Hospital in Izumo City, an annual meeting of the Japanese Primary Care Association at Osaka, the Japan Institute for Advancement of Medical Education Program in Tokyo, and the Japanese Consortium for General Medicine Teachers in Tokyo), and involved the first author explaining the same methodology each time, and the participants filling out the questionnaire directly. Inclusion criteria were as follows: possession of a Japanese medical license, practicing in a clinical setting, and willingness to provide informed consent for participation. Meanwhile, exclusion criteria were set as follows: non-physician profession, medical student, unwillingness to provide consent, and/or poor descriptions shared in responses (e.g., blank item responses).

### 2.2. Patient and Public Involvement

No patients were involved in the proceedings of this study.

### 2.3. Questionnaire Components

Demographic data, including age, gender, postgraduate year, specialty, practice settings, and hospital size, were first collected. Following previous research, diagnostic errors were defined as diagnoses that were delayed (e.g., clinically adequate and timely diagnoses were not made), wrong (e.g., before correct diagnoses were made), or completely missed (e.g., no diagnoses were ever made) [1].

Relevant detailed information on the most memorable diagnostic error cases was provided by the physicians themselves. This included situational information related to the occurrence of diagnostic errors (e.g., emergency visits or outpatient wards), time of the day that the error occurred, chief complaint, presence/absence of examination enforcement, initial and final diagnosis, presence/absence of specialists to consult within the hospital when an error in diagnosis occurred, type of cognitive bias, and type of error (e.g., missed, delayed, or incorrect diagnosis). Participants then reflected upon their own diagnostic error cases. We also inquired about the effects of 10 potential cognitive biases (i.e., anchoring bias, premature closure, availability bias, hassle bias, confirmation bias, overconfidence bias, visceral bias, base rate neglect bias, rule bias, and Maslow’s hammer, related to these error cases) and provided participants with explanations of these biases (Table 1).

Although more than 100 cognitive biases exist in the field of psychology, some were considered to overlap or deemed unimportant in the medical setting. The 10 biases listed in this study constitute representative cognitive biases based on previous research [22,23,24,25].

Finally, participants were asked to analyze any causative factors related to their recollections of diagnostic errors through subjective self-reflection. They were then asked to classify these into three categories (i.e., situational, data-gathering, and cognitive bias factors) and assign weights to each factor (they were informed that the weights should sum to 10; e.g., situational 4, data-gathering 3, and cognitive bias 3) [26].

### 2.4. Statistical Analyses

Categorical variables are presented as numbers and percentages, whereas continuous variables are presented as medians and interquartile ranges (IQR). The chi-squared test or the Fisher’s exact test was used to compare nominal variables. For continuous variables, *t*-tests or Wilcoxon rank-sum tests were used, as appropriate. Multivariate linear regression analysis and multivariate logistic regression analysis were conducted to examine the factors influencing the development of cognitive biases. The criteria used for each explanatory variable were set as follows: (i) factors associated with cognitive biases in previous studies [5,6,7,12,16,17,18,19,20,21,22,23,24,25,26] and univariate screening; factors that were significant at the level of *p* < 0.05 based on univariate regression. Specifically, gender, residents at the location where the error occurred, internists, emergency room, night-time, and weekends/holidays were selected in the multiple linear regression analysis for cognitive biases and the multiple logistic regression analysis with and without cognitive bias. All *p*-values < 0.05 were considered statistically significant, and all analyses were conducted using Stata Ver. 14.0 (StataCorp. 2015. Stata Statistical Software: Release 14. College Station, TX, USA: StataCorp LP).

## 3. Results

A total of 147 participants were recruited; four poor response descriptions were excluded, thirteen did not provide consent. Consequently, 130 participants completed and returned the questionnaires (88.4% response rate). Table 2 shows the participants’ baseline characteristics. The median age was 45 (IQR = 33−57), and 16.2% were women. Further, 55 participants specialized in internal medicine (43.0%), followed by 26 (20.3%) residents, and 21 (16.4%) family/general practitioners. In the context of this study, a resident is a doctor in training and in their first or second year of postgraduate training. In the sample, 47 (37.9%) participants worked in private clinics, while 53 (42.7%) worked in large hospitals (301 beds or more), 20 (16.1%) worked in medium-sized hospitals (101–300 beds), and four (3.2%) worked in small hospitals (less than 100 beds). The most common responses for the self-reported frequency of diagnostic errors in clinical practice were several times per month (31.8%), followed by several times each week (26.2%), and almost daily (11.1%).

Table 3 provides information on the most memorable diagnostic error cases described by participants in detail. The median number of years from medical school graduation to the time of the described diagnostic error case was four (IQR 2−15), while the emergency department was the most common error setting (61, 47.7%), followed by general outpatient clinics (37, 28.9%) and wards (18, 14.1%).

The errors were most common at night-time (52, 43.7%); regarding the day of the week, most errors (92, 78.6%) occurred on weekdays, when relatively large numbers of patients were visiting or staying in the clinical settings. However, approximately 21% of the most memorable diagnostic errors occurred on Saturdays, Sundays, and national holidays. Among these cases, 40% were identified within hours on the same day, and 8.7% required a few months to identify, but some cases required a few years.

After participants reviewed their diagnostic error cases, we asked them about the potential effects of the 10 cognitive biases on both the diagnostic processes and the errors. We found that a mean of 3.08 (*SD* = 1.48, min = 0, max = 8) cognitive biases were attributed to each diagnostic error case. Among these, anchoring bias (78, 60.0%) was the most frequent cause, followed by premature closure (76, 58.5%), availability bias (60, 46.2%), and hassle bias (43, 33.1%; Table 4).

Table 5 shows the top 10 final diagnoses among all the most memorable diagnostic error cases collected; relatively common but potentially fatal diseases (e.g., aortic dissection, aortic aneurysms, malignant tumors, and acute coronary syndrome) were included in the most remarkable cases.

We also classified the causative factors of the diagnostic errors into three categories: cognitive bias was considered the most common contributing factor (mean = 3.37, 95% CI 2.99−3.76), followed by data-gathering factors (mean = 2.73, 95% CI 2.42−3.04) and situational factors (mean = 2.66, 95% CI 2.31−3.01). Among participants who stated that cognitive bias contributed to their diagnostic errors, 111 (85.4%) reported that it affected 10% or more of such cases, 102 (78.5%) reported 20% or more, 81 (62.3%) reported 30% or more, and 63 (48.5%) reported 40% or more.

Based on the univariate and multivariate linear analyses, Table 6 shows the background variables that trigger cognitive bias for the most vividly recalled diagnostic errors. Analyses were performed for all demographic factors, but only gender and 1–2 years following postgraduation were statistically significant in the univariate analysis. Other factors that were indicated as possibly clinically associated with cognitive bias from previous quantitative and qualitative studies were included.

When the six variables were included, the results revealed a statistical predominance of emergency room care as a factor leading to cognitive bias (coefficient 1.04, *p*-value = 0.03), whereas other factors were not statistically significant. We also performed a multivariate logistic regression analysis for cognitive bias using the same variables and found that the value of emergency room care was statistically significant (adjusted odds ratio [aOR] 3.96, 95% CI 1.16−13.6, *p*-value = 0.028). In analysis, with factors associated with cognitive biases contributing to the most memorable diagnostic error cases extracted from the participants, the results of the multivariate linear analysis reconfirmed that emergency care settings were statistically predominantly correlated (coefficient 1.04, *p*-value = 0.03). Multivariate logistic regression analysis on the involvement of cognitive bias using the same variables also showed similar results (aOR 3.96, 95% CI 1.16–13.6, *p*-value = 0.028). Notably, participants who reflected on their errors encountered in the first two years following postgraduation were inversely correlated with cognitive bias as a cause of the diagnostic error (aOR 0.12, 95% CI 0.02 to 0.58, *p*-value = 0.009). Finally, we examined each of the 10 cognitive biases used in this study through a multivariate logistic analysis using the same six background factors as above. The results showed statistically significant correlations between errors encountered in the first two years following postgraduation and two cognitive biases, namely, overconfidence bias (aOR 4.08, 95% CI 1.35 to 12.30, *p*-value = 0.013) and hassle bias (aOR 4.45, 95% CI 1.47 to 13.50, *p*-value = 0.008).

## 4. Discussion

This study investigated the influence of cognitive bias on diagnostic errors by asking physicians to recall their most vividly memorable errors in a clinical setting. It also examined how often the physicians encountered diagnostic errors (i.e., the frequency of errors) in their current settings. From their self-reflections, we found that about 80% of the physicians encountered diagnostic errors at least a few times per month. This is comparable to data from previous studies, where the incidence of diagnostic errors among primary care physicians was 5–10% [12,27]. In this study, we attempted to include as much rigorous and detailed background and contextual information as possible. However, if a physician who reflected on their own diagnostic error cases uses the following recent definition noted in Improving Diagnosis in Health Care, the frequency of diagnostic errors may vary widely: “failure to establish an accurate and timely explanation of the patient’s health problem or communicate that explanation to the patient” [3].

Several reports show that the incidence rates of diagnostic errors in both radiology and dermatology settings (where the diagnostic environments are constant and diagnoses are made using visual systems) are lower than those in the primary care setting, which is the work setting of this study’s participants [28]. This may be due to the difficulty associated with providing proper diagnoses when conditions are uncertain; further, the primary care setting involves complex tasks and requires multitasking for undiagnosed first-visit patients [29,30,31]. Additionally, primary care settings have limited human and infrastructure resources and, owing to the need to treat patients with multimorbidity, physicians find it difficult to arrive at a correct diagnosis each time. However, in a highly specialized outpatient setting, specialist physicians tend to see a referred patient with a certain diagnosis, or with a differential diagnosis that is already narrowed down.

Although this study supports previous research findings demonstrating that cognitive bias is the most significant cause of diagnostic error in the medical setting [14,16,29], situational and data-gathering factors also substantially contribute to such errors. Situational factors include being too busy, not being able to sleep at night, having too many patients, and lack of access to consultation. The most memorable diagnostic error cases are relatively often from the weekends and in holiday or night situations, for which there are two possible reasons. Japan has the highest number of hospital beds and outpatients per physician and number of hospitals per patient in the world [32]. Consequently, the examination and medical support systems are often inadequate on weekends and holidays. On the other hand, Japan’s medical system follows a free-access model, allowing patients to see a doctor freely [32]. For this reason, hospitals with a high concentration of patients are often staffed by relatively young physicians, who often practice alone with little manpower. It is thus likely that the diagnostic errors that occur in such difficult environments lead to the most memorable diagnostic error cases.

Meanwhile, the most common data-gathering factors were failure to carefully conduct physical exams and not performing the necessary tests in the context of actual clinical practice [19]. Notably, residents cited a lack of knowledge and experience as the most significant factors, rather than cognitive bias. Further, they understandably tended to implicate data-gathering and situational factors as reasons for such errors.

One study found that about half of the errors involved both system and cognitive errors [1]. Cognitive biases were the sole reason for about 30% of the errors, and system errors were the primary reason for about 20% of the errors; no-fault factors accounted for 7% of the errors [1]. We believe that the analysis of causes of diagnostic errors, including that in our study, depends largely on the proficiency of the participating physicians and their specialty. For instance, when we considered the final diagnoses observed by participants, the majority were common diseases. We required participants to use the “most vividly recallable cases” because we predicted that they might list rare diseases for their reflection; however, the outcome did not conform to this prediction. In other words, most of the cases reported by participants were either a common disease or a rare presentation of common diseases. Since we analyzed the diagnosis error cases that they recalled most vividly, we were more likely to see error cases accompanied by painful or sad emotions rather than those regarding diseases that were challenging to diagnose. The association of cognitive bias was also likely to be different in non-fatal cases, since error cases that did not have more severe outcomes were less likely to be included. We believe that primary care physicians, with some experience, are less likely to exhibit a lack of knowledge regarding common diseases. On the contrary, in the first two years following postgraduation, physicians often lack the experience, basic knowledge, and diagnostic skills to accurately collect information on common diseases. Previous research suggests that this is because such factors are more likely than cognitive biases to cause problems when physicians have insufficient experience or knowledge (e.g., among students and residents) [25].

Among the 10 representative cognitive biases we presented to participants, the top three were anchoring bias, premature closure, and availability bias. This finding was similar to those of previous studies with emergency room physicians and internists [16,22,23,24,33,34]. Interestingly, after adjusting for the multivariate analysis, we found that working in an emergency room as an emergency physician significantly induced cognitive bias due to two reasons. First, few emergency physicians in Japan are certified by the Japan Professional Medical Association. In fact, general surgeons and physicians must often work as temporary emergency physicians at night and on weekends/holidays. Second, emergency room care requires more complex thinking and simultaneous multitasking; hence, the time to conduct diagnoses is limited [17,35,36].

### Limitations

Some limitations of this study need to be acknowledged. The first was selection bias due to the sampling methodology adopted; as our participants were recruited at conferences and meetings for internal medicine, no other specialties were included. Furthermore, even if a participant reported being an internal medicine and family/general practitioner during recruitment, this may not be accurate, as some surgeons may have switched to internal medicine or general practice. The participants who are willing to learn in these clinical reasoning and diagnostic error workshops are selective and likely to be more passionate about the subject than the average physician, and it is questionable whether the results of this study are applicable to other types of physicians. However, we could adjust for the widely varying frequency of cognitive biases and environmental factors depending on the participants, and we believe that this provides a partial solution, although not a perfect one. Additionally, we obtained detailed information about diagnostic errors through a high response rate. The second limitation concerned recall bias, which implies that participants’ memories may not have been entirely accurate. However, this was an unavoidable limitation due to the self-reflective nature of the survey questionnaire. To extract vivid details, we also required diagnostic error cases in which participants could best recall the circumstances and their psychological states at the time. Third, a possible weakness may be related to external validity. In other words, diagnostic errors are greatly influenced by the context of clinical practice. In this regard, participants may have specifically been influenced by the Japanese health care system and education, in which a few physicians are responsible for many beds and outpatients [32]. Thus, it may be difficult to accurately compare the causes of cognitive bias reported in this study to those of physicians working in other countries. The fourth limitation was the representativeness of the cognitive biases that we considered. Specifically, we listed 10 representative biases following those discussed in related research [22,23,24]. However, other cognitive biases may result in more possible effects in actual clinical practice. It is highly impractical to enumerate all such biases in the context of one survey study. Further, every cognitive bias often overlaps as a cause; therefore, it may be difficult to rigorously categorize the diagnostic processes used by physicians.

## 5. Conclusions

The results of this study revealed that cognitive bias was involved in the most common diagnostic errors of physicians in Japan. However, situational and data-gathering factors also contributed substantially to diagnostic errors. Among the 10 representative cognitive biases presented to participants, anchoring bias, premature closure, and availability bias were the most common. Further, the emergency room care situation may also be a factor in cognitive bias leading to diagnostic errors. To reduce the risk of diagnostic errors, physicians should know about the potential risk conditions of diagnostic processes.

## Figures and Tables

**Table 1 ijerph-19-04645-t001:** List of representative cognitive biases shared with participants.

Cognitive Biases	Explanation
Availability bias	It is instinctive to think of things that come to mind easily. This is also influenced by what I have been through recently, among other things.
Overconfidence bias	It is easy to believe judgements about the self and others who are overconfident.
Anchoring bias	You cling on to your first thought and do not consider other possibilities.
Confirmation bias	This entails underestimating information that does not fit one’s hypothesis.
Hassle bias	This is associated with quickly processing thoughts that we physically and mentally process with ease.
Rule bias	You blindly follow general rules that are not always correct.
Base rate neglect	Individuals may ignore the frequency of a disease; sometimes, finding a rare disease accelerates this even further.
Visceral bias	You may have positive or negative feelings about the patient, which may influence your decision.
Premature closure	Upon making a diagnosis, you cease to think about it further. This is a strong bias that may contribute most to errors.
Maslow’s hammer	When using a hammer, you want to hit a nail. This is easier to do when you have a technique (e.g., endoscopy and cardiac catheterization).

**Table 2 ijerph-19-04645-t002:** Baseline characteristics of survey participants.

**Age** (Median)		***n* = 130**
		45 (IQR 33−57)
**Gender**		***n* = 130**
	Female	21 (16.2%)
**Postgraduate year** (Median)		***n* = 130**
		18 (5−30)
**Specialty**		***n* = 128**
	Internal medicine	55 (43.0%)
	Resident	26 (20.3%)
	Family practice and general practice	21 (16.4%)
	Others	26 (20.3%)
**Facility**		***n* = 124**
	Clinic	47 (37.9%)
	Small hospital (20−100 beds)	4 (3.2%)
	Medium-sized hospital (101−300 beds)	20 (16.1%)
	Large/university hospital (301 beds and up)	53 (42.7%)
**Frequency of diagnostic errors**		***n* = 126**
	Almost daily	14 (11.1%)
	Several times each week	33 (26.2%)
	Once per week	13 (10.3%)
	Several times each month	40 (31.8%)
	Once per month	10 (7.9%)
	Once every few months	4 (3.2%)
	Biannually	9 (7.1%)
	Annually	3 (2.4%)

**Table 3 ijerph-19-04645-t003:** Analysis of the most memorable diagnostic error cases reported by participants.

**Postgraduate year when case was encountered**	***n* = 127**	4 (IQR 2−15)
**Case location**	***n* = 128**	
Emergency room		61 (47.7%)
General outpatient office		37 (28.9%)
Ward		18 (14.1%)
Other		6 (4.7%)
Specialist outpatient office		4 (3.1%)
Operating room		2 (1.6%)
**Working hours when case was encountered**	***n* = 119**	
Morning (08:30−12:00)		29 (24.4%)
Afternoon (13:00−17:00)		34 (28.6%)
Night (17:00−08:30)		52 (43.7%)
Other		4 (3.4%)
**Day of the week**	***n* = 117**	
Monday−Friday		92 (78.6%)
Saturday, Sunday/Holiday		25 (21.4%)
**Time taken for detecting the diagnostic error**	***n* = 126**	
Within 60 min		12 (9.5%)
Within a few hours		40 (31.7%)
Within a few days		43 (34.1%)
Within a few weeks		18 (14.3%)
Within a few months		11 (8.7%)
Within a few years		2 (1.6%)

**Table 4 ijerph-19-04645-t004:** Cognitive biases in diagnostic error cases that could be most vividly recalled and described.

How Many of the 10 Representative Biases Impacted the Case?	Mean = 3.08 (*SD* = 1.48: Min 0, Max 8)
10 representative biases	*n* = 130
Anchoring bias	78 (60%)
Premature closure	76 (58.5%)
Availability bias	60 (46.2%)
Hassle bias	43 (33.1%)
Confirmation bias	42 (32.3%)
Overconfidence bias	41 (31.5%)
Visceral bias	26 (20%)
Base rate neglect bias	9 (6.9%)
Rule bias	8 (6.2%)
Maslow’s hammer	6 (4.6%)

**Table 5 ijerph-19-04645-t005:** Top 10 final diagnoses among all the most memorable diagnostic error cases collected (*n* = 127).

No.	Type	Count	Percent
1	Thoracoabdominal vascular disease	15	11.5
2	Cancer	13	10.0
3	Acute coronary syndrome (ACS)	8	6.2
4	Appendicitis	8	6.2
5	Peritonitis	8	6.2
6	Cardiovascular disease (except for ACS)	7	5.4
7	Stroke	6	4.6
8	Metabolic encephalopathy	5	3.8
9	Central nervous system (CNS) infections	5	3.8
10	Skin, bone, soft tissue infection	5	3.8

**Table 6 ijerph-19-04645-t006:** Contribution to cognitive bias based on univariate and multivariate analysis.

	Cognitive Biases Contribution (0–10)				Contributed Cognitive Bias (Yes or No)			
	Simple linear regression analysis for cognitive bias contribution level		Multiple linear regression analysis for cognitive bias contribution level		Univariate logistic analysis for with or without cognitive bias contributing		Multiple logistic analysis for with or without cognitive bias contributing	
	Coefficient (95% CI)	*p*-value	Coefficient (95% CI)	*p*-value	OR (95% CI)	*p*-value	aOR (95% CI)	*p*-value
Male	0.29 (−0.75 to 1.34)	0.579	0.56 (−0.49 to 1.61)	0.290	5.63 (1.11 to 28.4)	**0.036**	1.34 (0.32 to 5.58)	0.683
1–2 years following postgraduation	−1.0 (−1.94 to −0.50)	**0.039**	−0.86 (−1.98 to 0.24)	0.123	0.35 (0.12 to 1.02)	0.054	0.12 (0.02 to 0.58)	**0.009**
Internal medicine	0.72 (−0.44 to 1.49)	0.064	0.46 (−0.39 to 1.30)	0.281	1.30 (0.48 to 3.57)	0.602	0.75 (0.21 to 2.67)	0.652
Emergency room	0.62 (−0.27 to 1.51)	0.169	1.04 (0.12 to 1.96)	**0.027**	2.63 (0.94 to 7.29)	0.062	3.94 (1.15 to 13.51)	**0.029**
Night shift	−0.65 (−1.43 to 0.12)	0.098	−0.55 (−1.38 to 0.30)	0.208	1.53 (0.54 to 4.33)	0.42	2.00 (0.56 to 7.32)	0.280
Weekends and holidays	−0.36 (−1.33 to 0.61)	0.462	−0.34 (−1.33 to 0.65)	0.500	1.32 (0.35 to 4.93)	0.41	1.32 (0.32 to 5.49)	0.698

Significant *p*-values are presented in bold. Coefficient and adjusted odds ratios (aOR) and 95% confidence intervals (CI) are reported. The variables male, 1−2 years following postgraduation, internal medicine, emergency room, night shift, and weekends and holidays are incorporated in the multiple linear regression analysis for cognitive biases’ proportion (0–10) and the multiple logistic regression analysis with and without cognitive bias contributing.

## Data Availability

The data that support the findings of this study are available from the General Medicine Center, Shimane University Hospital, upon reasonable request.

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
