# Peer review of "Cognitive Bias and Diagnostic Errors among Physicians in Japan: A Self-Reflection Survey"

_ijerph, 2022, doi:10.3390/ijerph19084645_

Round 1
Reviewer 1 Report
In this cross-sectional study, authors developed a 19-item questionnaire to investigate the frequency of cognitive biases and diagnostic errors.
The article is well organized and structured, and results are clearly described in tables.
Minor changes could improve the quality and the clarity of the study:
- Authors did not include the evaluation of the severity of the diagnostic errors, classifying their consequences into multiple categories (e.g. no consequences, minor consequences, major consequences, fatal)
- Table 1 – 2 – 3 – 4: it is not necessary to use bold type for each value, but only for the most general ones (e.g. n = …)
- Table 5: authors should use the bold type to highlight statistically significant p-values
- Page 7 Line 218: the citation should be in the right style
Author Response
We would like to re-submit the manuscript titled “Cognitive bias and diagnostic errors among physicians in Japan: a self-reflection survey.”
We have carefully rechecked this manuscript and made appropriate changes in accordance with the reviewers’ suggestions. The point-by-point responses to their comments have been prepared and are presented below.
We thank you and the reviewers for the thoughtful suggestions and insights, which have enriched the manuscript and produced a more balanced and better account of our research. We hope that the revised manuscript is now suitable for publication in your journal.
We look forward to your response.
Sincerely,
Takashi Watari, MD, MHQS, PhD
89-1, Enya-cho, General Medicine Centre
Shimane University Hospital, Izumo shi
Shimane, Japan 693-8501.
Phone: +81-853-20-2005
Fax: +81-853-20-2375
E-mail: wataritari@gmail.com
Response: Thank you for your helpful and constructive recommendation. We have made the following changes as per your suggestions. Tables 1, 2, 3, 4, and 5 have been modified, as shown in the revised manuscript. Moreover, in Page 7 Line 218, we have included “One study found that about half of the errors involved both system and cognitive errors [1].”
Thank you for pointing this out to us: "Authors did not include an evaluation of the severity of the diagnostic errors, classifying their consequences into multiple categories” We originally tried to elucidate the outcomes that the reviewer mentioned, but the Ethics Committee cautioned us that they were psychologically burdensome for participants, thus we removed them. As a result, the final outcome for patients with diagnostic errors is unknown. Thank you for your constructive feedback.
Reviewer 2 Report
Line 17. The Authors wrote 7 conferences, but they mentioned only 6 ones.
Line 25. Replace “.028” with “0.028”.
Lines 30-54. I believe that the number of references reported in this section is very limited.
Lines 94-95. Please rephrase this sentence because it is unclear in my opinion.
Lines 113-115. Please replace “.05” with “0.05”.
Line 137. In my opinion, the Authors should better explain when a diagnostic error is considered as memorable error.
Line 143. I do not understand why the Authors grouped the days of the week in Mon-Thu, Fri, Sat, and Sunday/holiday. Is there a specific reason for this selection?
Lines 177-180-182. Please see my comments in Lines 25 and 113-115.
Line 182. Observe that “Postgraduate year 1-2” was also statistically significant considering the multiple logistic analysis. Please comment also on this result.
Line 266. Why are you including also “religious and cultural differences”? Please clarify this aspect.
Author Response
We would like to re-submit the manuscript titled “Cognitive bias and diagnostic errors among physicians in Japan: a self-reflection survey.”
We have carefully rechecked this manuscript and made appropriate changes in accordance with the reviewers’ suggestions. The point-by-point responses to their comments have been prepared and are presented below.
We thank you and the reviewers for the thoughtful suggestions and insights, which have enriched the manuscript and produced a more balanced and better account of our research. We hope that the revised manuscript is now suitable for publication in your journal.
We look forward to your response.
Sincerely,
Takashi Watari, MD, MHQS, PhD
89-1, Enya-cho, General Medicine Centre
Shimane University Hospital, Izumo shi
Shimane, Japan 693-8501.
Phone: +81-853-20-2005
Fax: +81-853-20-2375
E-mail: wataritari@gmail.com
- Line 17. The Authors wrote 7 conferences, but they mentioned only 6 ones.
Response: Thank you. We have corrected the same as per your suggestions.
“A self-reflection questionnaire survey on physicians’ most memorable diagnostic error cases was conducted at seven conferences: one each in Okayama, Hiroshima, Matsue, Izumo City, Osaka, and two in Tokyo.”
Line 25. Replace “.028” with “0.028”.
Response: We have corrected all the related mentions. Thank you
Lines 30-54. I believe that the number of references reported in this section is very limited. 1
Response: Thank you very much. We agree; we went through the section once again and realized that some of the literature was outdated. We have presently created a new background section.
Lines 94-95. Please rephrase this sentence because it is unclear in my opinion.
Response: We apologize for the lack of clarity in the statement. We had our manuscript rechecked by an English language editing service and have implemented the necessary changes for clarity and language. Thank you for your constructive feedback.
P3 L105
“Participants then reflected upon their own diagnostic error cases. We also inquired about the effects of 10 potential cognitive biases (i.e., anchoring bias, premature closure, availability bias, hassle bias, confirmation bias, overconfidence bias, visceral bias, base rate neglect bias, rule bias, and Maslow’s hammer, related to these error cases) and provided participants with explanations of these biases (Table 1).”
Lines 113-115. Please replace “.05” with “0.05”.
Response: We have corrected all the related mentions in the revised paper. Thank you.
Line 137. In my opinion, the Authors should better explain when a diagnostic error is considered as memorable error.
Response: Thank you for your appropriate remark. We have now added a more detailed explanation of the definition.
P2, L60
“To facilitate participant comprehension, we also provided them with a document explaining how to define diagnostic errors, classify cognitive biases, and analyze the situational factors described later. Participants filled out the survey themselves; the survey was used to collect information on cases defined as the most vividly recalled cases of clear diagnostic errors in a physician’s career, while obtaining detailed situational and environmental information related to the same.,”
Line 143. I do not understand why the Authors grouped the days of the week in Mon-Thu, Fri, Sat, and Sunday/holiday. Is there a specific reason for this selection?
Response: Thank you for your helpful and constructive feedback. We agree with your point. Indeed, Friday will be our normal operation day. Previous surveys have shown that patients are often referred to acute care hospitals before Saturday and Sunday. We initially distinguished Saturday outpatient clinics and other operations separately, as those vary according to their facility management style.
However, given this context, there is little rationale in separating them in detail and there is a high risk of confusing the reader; thus, we have reclassified them into two categories: Weekday and Weekend.
“Day of the week
Monday-Friday n=92 (78.6%)
Saturday, Sunday/Holiday n=25 (21.4%)”
Lines 177-180-182. Please see my comments in Lines 25 and 113-115.
Response: We apologize for our oversight. We have corrected all p-value descriptions as the reviewer indicated.
Line 182. Observe that “Postgraduate year 1-2” was also statistically significant considering the multiple logistic analysis. Please comment also on this result.
Response: Thank you for this important point. I fully agree with you. To address the same, we have added our interpretation to the Conclusions and Discussion sections.
P7 L203
“Notably, participants who reflected on their errors encountered in the first two years following postgraduation were inversely correlated with cognitive bias as a cause of the diagnostic error (aOR 0.12, 95% CI 0.02 to 0.58, p-value = 0.009).”
P9 L271
“On the contrary, in the first two years following postgraduation, physicians often lack the experience, basic knowledge, and diagnostic skills to accurately collect information on common diseases. Previous research suggests that this is because such factors are more likely than cognitive biases to cause problems when physicians have insufficient experience or knowledge (e.g., among students and residents) [25].”
Line 266. Why are you including also “religious and cultural differences”? Please clarify this aspect.
Response: We think that you have raised an accurate point here. We apologize for the inadequate explanation. To clarify, Japanese culture is quite unique, represented by phenomena such as HARAKIRI suicide, which is considered to be responsible, and it is certainly difficult to provide a single-sentence explanation on this matter. It is also difficult to explain cultural aspects quantitatively; thus, we have removed these details as they lack scientific validity. Thank you.
P10 L299
“Third, a possible weakness may be related to external validity. In other words, diagnostic errors are greatly influenced by the context of clinical practice. In this regard, participants may have specifically been influenced by the Japanese health care system and education, in which a few physicians are responsible for many beds and outpatients [32]. Thus, it may be difficult to accurately compare the causes of cognitive bias reported in this study to those of physicians working in other countries.”
Reviewer 3 Report
I enjoyed reading this manuscript. Here are some concerns that I would recommend the Authors keep in mind when revising the manuscript:
1) Methods: I don't agree with the assertion that convenience sampling ensures participant diversity. If you really want to ensure diversity (i.e. population representativeness), you need to plan a survey, give appropriate weights, and make your calculations by applying those weights. By sticking to a planned survey scheme, you can also assess whether the sample you end up having is representative or not. Convenience sampling is, well, convenient, but it's always methodologically weak.
2) In general, given the way the study sample was put together, the generalizability of results is a concern, and it should be clearly acknowledged, in the Discussion (more than it is right now) and possibly in the Abstract as well.
3) It is not clear what variable is used as the dependent variable in the linear and logistic regression models. Is the dependent variable of the former obtained by splitting the dependent variable of the former into two categories? Please clarify.
4) Another major concern I have regards the definition of "most memorable". Were participating physicians given any explanation of what that means? E.g. link wth severity, rarity, shame that they felt, etc. If not, that is a major limitation as different physicians may interpret very differently that expression. For instance, I see that the "most memorable error" occurred a median 4 years from graduation, and this closeness to one physician's early career years may be caused by the definition. From Table 10, I see that the reported "most memorable errors" were mostly related to severe health conditions (or health conditions that may have very bad consequences if misdiagnosed), while I would have defined a "most memorable error" as an error where the diagnosis could have been easy if the physician had avoided cognitive biases (regardless of severity). In general, a lack of guide of what "most memorable" should mean may have affected the results, and this should be acknowledged as an important limitation.
5) Results: "most errors (59.8%) occurred on weekdays", but weekdays are the majority of days! It actually looks like the number of errors per day is higher on weekends and holidays... is that true? Or am I mistaken somewhere? Please clarify. This also has consequences on the models that were fitted.
6) Results: the significant result (in the logistic analysis) of the variable "postgraduate year 1-2" is not mentioned. Why? Don't the Authors trust that result? Anyway, since they included the variable in the model and presented in the table, it should be commented in the text (Results and Discussion).
Author Response
We would like to re-submit the manuscript titled “Cognitive bias and diagnostic errors among physicians in Japan: a self-reflection survey.”
We have carefully rechecked this manuscript and made appropriate changes in accordance with the reviewers’ suggestions. The point-by-point responses to their comments have been prepared and are presented below.
We thank you and the reviewers for the thoughtful suggestions and insights, which have enriched the manuscript and produced a more balanced and better account of our research. We hope that the revised manuscript is now suitable for publication in your journal.
We look forward to your response.
Sincerely,
Takashi Watari, MD, MHQS, PhD
89-1, Enya-cho, General Medicine Centre
Shimane University Hospital, Izumo shi
Shimane, Japan 693-8501.
Phone: +81-853-20-2005
Fax: +81-853-20-2375
E-mail: wataritari@gmail.com
3. 1) Methods: I don't agree with the assertion that convenience sampling ensures participant diversity. If you really want to ensure diversity (i.e. population representativeness), you need to plan a survey, give appropriate weights, and make your calculations by applying those weights. By sticking to a planned survey scheme, you can also assess whether the sample you end up having is representative or not. Convenience sampling is, well, convenient, but it's always methodologically weak.
Response: Thank you very much for your constructive feedback. It seems that when we translated the text into English, the wording was different. We have now made a careful correction. We have also added a limitation on the methodological weakness of our study. Thank you very much for your suggestions. We sincerely thank the reviewers for the valuable time invested to review our paper.
P2L69
“This paper-based data collection process was conducted immediately before the workshop at the place or hospital where the participants worked. Participant recruitment was conducted at conferences in seven different areas of Japan (i.e., Okayama University Hospital in Okayama, Hiroshima City Hospital in Hiroshima, Matsue Seikyo Hospital in Matsue, Shimane University Hospital in Izumo City, an annual meeting of the Japanese Primary Care Association at Osaka, the Japan Institute for Advancement of Medical Education Program in Tokyo, and the Japanese Consortium for General Medicine Teachers in Tokyo), and involved the first author explaining the same methodology each time, and the participants filling out the questionnaire directly.”
P9L292
“The participants who are willing to learn in these clinical reasoning and diagnostic error workshops are selective and likely to be more passionate about the subject than the average physician, and it is questionable whether this can be applied to other type of physicians.”
2) In general, given the way the study sample was put together, the generalizability of results is a concern, and it should be clearly acknowledged, in the Discussion (more than it is right now) and possibly in the Abstract as well.
Response: Thank you for your pertinent points. We had attempted to clearly describe the information, but it was insufficient. We agree with the reviewer's opinion. We have created the Limitation section more clearly at present. We also added one line related to this point to the abstract.
P9, L286
“Some limitations of this study need to be acknowledged. The first was selection bias due to the sampling methodology adopted; as our participants were recruited at conferences and meetings for internal medicine, no other specialties were included. Furthermore, even if a participant reported being an internal medicine and family/general practitioner during recruitment,, this may not be accurate, as some surgeons may have switched to internal medicine or general practice. The participants who are willing to learn in these clinical reasoning and diagnostic error workshops are selective and likely to be more passionate about the subjects than the average physician, and it is questionable whether this can be applied to other type of physicians. However, we could adjust for the widely varying frequency of cognitive biases and environmental factors depending on the participants, and we believe that this provides a partial solution, although not a perfect one.”
P1L23
“Although limited to a certain extent by its sample collection, due to the sensitive nature of information regarding physicians’ diagnostic errors, this study nonetheless shows correlations with environmental factors (emergency room care situations) that induce cognitive biases that cause diagnostic errors.”
3) It is not clear what variable is used as the dependent variable in the linear and logistic regression models. Is the dependent variable of the former obtained by splitting the dependent variable of the former into two categories? Please clarify.
Response: We apologize for the misleading explanation. We have corrected the text as follows and added a note to the Method section and Table 6. We thank the reviewers for their constructive review.
Method
“They were then asked to classify these into three categories (i.e., situational, da-ta-gathering, and cognitive bias factors) and assign weights to each factor (they were informed that the weights should sum up to 10; e.g., situational 4, data-gathering 3, and cognitive bias 3)”
“Multivariate linear regression analysis and multivariate logistic regression analysis were conducted to examine the factors influencing the development of cognitive biases. The criteria used for each explanatory variable were set as follows: i) factors associated with cognitive biases in previous studies [5–8, 16–19] and ii) univariate screening; factors that were significant at the level of p < 0.05 based on univariate regression. Specifically, gender, residents, age, internists, emergency room, night-time, and weekends/holidays were selected in the multiple linear regression analysis for cognitive biases and the multiple logistic regression analysis with and without cognitive bias.”
Table 6.
“Significant p-values are presented in bold. Coefficient and adjusted odds ratios (aOR) and 95% confidence intervals (CI) are reported. The variables male, 1−2 years following postgraduation, internal medicine, emergency room, night shift, and weekends and holidays are incorporated in the multiple linear regression analysis for cognitive biases proportion (0-10) and the multiple logistic regression analysis with and without cognitive bias contributing.”
4) Another major concern I have regards the definition of "most memorable". Were participating physicians given any explanation of what that means? E.g. link wth severity, rarity, shame that they felt, etc. If not, that is a major limitation as different physicians may interpret very differently that expression. For instance, I see that the "most memorable error" occurred a median 4 years from graduation, and this closeness to one physician's early career years may be caused by the definition. From Table 10, I see that the reported "most memorable errors" were mostly related to severe health conditions (or health conditions that may have very bad consequences if misdiagnosed), while I would have defined a "most memorable error" as an error where the diagnosis could have been easy if the physician had avoided cognitive biases (regardless of severity). In general, a lack of guide of what "most memorable" should mean may have affected the results, and this should be acknowledged as an important limitation.
Response: Thank you for the excellent feedback. We wholly agree with the reviewer’s points and that our description was misleading. We actually provided guidance, as described in the description, before the survey. We have now modified the description in the paper as follows:
“Participant recruitment was conducted at conferences in seven different areas of Japan (i.e., Okayama University Hospital in Okayama, Hiroshima City Hospital in Hiroshima, Matsue Seikyo Hospital in Matsue, Shimane University Hospital in Izumo City, an annual meeting of the Japanese Primary Care Association at Osaka, the Japan Institute for Advancement of Medical Education Program in Tokyo, and the Japanese Consortium for General Medicine Teachers in Tokyo), and involved the first author explaining the same methodology each time, and the participants filling out the questionnaire directly.”
“Participants filled out the survey themselves; the survey was used to collect information on cases defined as the most vividly recalled cases of clear diagnostic errors in a physician’s career, while obtaining detailed situational and environmental information related to the same.”
5) Results: "most errors (59.8%) occurred on weekdays", but weekdays are the majority of days! It actually looks like the number of errors per day is higher on weekends and holidays... is that true? Or am I mistaken somewhere? Please clarify. This also has consequences on the models that were fitted.
Response: Thank you for this important point. I apologize for the lack of explanation. Indeed, it was true, and this was also the case in our previous Japanese study. W4 added this note for several possible reasons. We have incorporated further details in the text to clarify the same, as follows:
P8, L242
“The most memorable diagnostic error cases are relatively often from the weekends and in holiday or night situations, for which there are two possible reasons. Japan has the highest number of hospital beds and outpatients per physician and number of hospitals per patient in the world [32]. Consequently, the examination and medical support systems are often inadequate on weekends and holidays. On the other hand, Japan's medical system follows a free-access model, allowing patients to see a doctor freely [32]. For this reason, hospitals with a high concentration of patients are often staffed by relatively young physicians, who often practice alone with little manpower. It is thus likely that the diagnostic errors that occur in such difficult environments lead to the most memorable diagnostic error cases.”
P6, L161
“The errors were most common at night time (52, 43.7%); regarding the day of the week, most errors (92, 78.6%) occurred on weekdays, when relatively large numbers of patients were visiting or staying in the clinical settings. However, approximately 21% of the most memorable diagnostic errors occurred on Saturdays, Sundays, and national holidays.”
6) Results: the significant result (in the logistic analysis) of the variable "postgraduate year 1-2" is not mentioned. Why? Don't the Authors trust that result? Anyway, since they included the variable in the model and presented in the table, it should be commented in the text (Results and Discussion).
Response: We apologize for our mistake in this regard. The result is still the same, but the part we mentioned was missing. Hence, we are adding it again. Thank you for pointing out this crucial error. We have also added the same in the Discussion section.
P7, L199
“Notably, participants who reflected on their errors encountered in the first two years following postgraduation were inversely correlated with cognitive bias as a cause of the diagnostic error (aOR 0.12, 95% CI 0.02 to 0.58, p-value = 0.009).”
P9. L271
“On the contrary, in the first two years following postgraduation, physicians often lack the experience, basic knowledge, and diagnostic skills to accurately collect information on common diseases. Previous research suggests that this is because such factors are more likely than cognitive biases to cause problems when physicians have insufficient experience or knowledge (e.g., among students and residents) [25].”
Reviewer 4 Report
Watari et al. have conducted a survey-based statistical study to assess cognitive biases associated with diagnostic errors in physicians in Japan. After statistically analyzing the 130 questionnaires they received back, they concluded that the most frequent cognitive biases were anchoring bias, premature closure and availability bias.
In general, the work is very well written, very clear and detailed. The methodology carried out seems correct and the results are analyzed and discussed correctly. Also note the fact that a "limitations" section has been included.
I would only make a small modification; and it is that, from my point of view, the table present in the Supplementary Material should be in the Main Document, since it is very enlightening for a reader who, like me, is not familiar with all the cognitive biases that are evaluated in this study.
Author Response
We would like to re-submit the manuscript titled “Cognitive bias and diagnostic errors among physicians in Japan: a self-reflection survey.”
We have carefully rechecked this manuscript and made appropriate changes in accordance with the reviewers’ suggestions. The point-by-point responses to their comments have been prepared and are presented below.
We thank you and the reviewers for the thoughtful suggestions and insights, which have enriched the manuscript and produced a more balanced and better account of our research. We hope that the revised manuscript is now suitable for publication in your journal.
We look forward to your response.
Sincerely,
Takashi Watari, MD, MHQS, PhD
89-1, Enya-cho, General Medicine Centre
Shimane University Hospital, Izumo shi
Shimane, Japan 693-8501.
Phone: +81-853-20-2005
Fax: +81-853-20-2375
E-mail: wataritari@gmail.com
Response: We are highly appreciative of your detailed and constructive feedback. We will consider your suggestion and introduce the Supplementary Material as Table 1. in the main document. Thank you very much.
Reviewer 5 Report
This is an interesting study on cognitive biases associated with diagnostic errors. It is a topic that will be of great interest to readers because any practitioner remembers diagnostic errors in their career as some of the most difficult moments in the professional life and will be interested in acquiring new information and insights on this topic. The introduction is appropriate and informative. The methods and results are set out very clearly. The discussion has an adequate number of references and provides interesting food for thought. The only relevant methodological limitation, but already pointed out by the authors, is that the recruitment of participants took place only at meetings of internal medicine, resulting in a selection of only internists, whereas it would have been very interesting to be able to make comparisons between different specialties. Perhaps the authors may consider replicating the study design in the future and including populations of different specialists. English should be checked before publication.
Author Response
We would like to re-submit the manuscript titled “Cognitive bias and diagnostic errors among physicians in Japan: a self-reflection survey.”
We have carefully rechecked this manuscript and made appropriate changes in accordance with the reviewers’ suggestions. The point-by-point responses to their comments have been prepared and are presented below.
We thank you and the reviewers for the thoughtful suggestions and insights, which have enriched the manuscript and produced a more balanced and better account of our research. We hope that the revised manuscript is now suitable for publication in your journal.
We look forward to your response.
Sincerely,
Takashi Watari, MD, MHQS, PhD
89-1, Enya-cho, General Medicine Centre
Shimane University Hospital, Izumo shi
Shimane, Japan 693-8501.
Phone: +81-853-20-2005
Fax: +81-853-20-2375
E-mail: wataritari@gmail.com
Response: We sincerely thank you for your kind compliments. We will seriously consider working on reducing adverse events caused by medical errors through further research on diagnostic errors. We thank you for investing your valuable time to review our manuscript.
Reviewer 6 Report
Authors investigated the characteristics of cognitive biases related to
with diagnostic errors to clarify how cognitive biases and situational factors related to diagnostic errors. There are several major issues to be published.
1. The author should need to calculate the sample size in the cross sectional study.
2. The author should summarize information about most often case for diagnostic errors, etc. such as Correct Diagnosis What Went Wrong? Why Did It Happen? etc. It is insufficient information to count the diagnostic errors.
3. In the statistical analysis part, the authors should described all analysis methods, such as what kind of statistical test the authors use.
4. The authors should write the variable selection method for the multivariate analysis. The authors described only multivariate analysis results, but the univariate analysis should be also described.
5. The authors counted about any bias, but the authors need to analyze the association study if each bias. In addition, the authors should analyze further about the characteristics of physicians who have many errors.
6. The authors should tabulate about the recommendation/solution to avoid each medical bias.
Author Response
We would like to re-submit the manuscript titled “Cognitive bias and diagnostic errors among physicians in Japan: a self-reflection survey.”
We have carefully rechecked this manuscript and made appropriate changes in accordance with the reviewers’ suggestions. The point-by-point responses to their comments have been prepared and are presented below.
We thank you and the reviewers for the thoughtful suggestions and insights, which have enriched the manuscript and produced a more balanced and better account of our research. We hope that the revised manuscript is now suitable for publication in your journal.
We look forward to your response.
Sincerely,
Takashi Watari, MD, MHQS, PhD
89-1, Enya-cho, General Medicine Centre
Shimane University Hospital, Izumo shi
Shimane, Japan 693-8501.
Phone: +81-853-20-2005
Fax: +81-853-20-2375
E-mail: wataritari@gmail.com
# 1. The author should need to calculate the sample size in the cross sectional study.
Response: Thank you for pointing this out to us. We have included our sample size calculations as part of our study protocol.
P2, L69
“Regarding sample size, we assumed that cognitive bias was involved in 50% of diagnostic errors that occurred in the emergency room, a challenging environment, and that 25% of diagnostic errors occurred elsewhere. For a desired power of 0.80 and alpha 0.05, we estimated that we would need 124 participants to detect the differences. A total of 155 participants were considered as the target sample size, excluding 15% of the participants who disagreed to engage in the study and 5% of the participants who answered inappropriately.”
- The author should summarize information about most often case for diagnostic errors, etc. such as Correct Diagnosis What Went Wrong? Why Did It Happen? etc. It is insufficient information to count the diagnostic errors.
Response: Thank you very much for your time and attention.
The purpose of this study was to examine the relationship between cognitive bias and diagnostic error cases. The question regarding how we can improve this (which, as you pointed out, is what many physicians would like to know the most) is outside the scope of the survey and we did not collect information on this topic. We are correcting various contents of the text as issues to be addressed in the future. Thank you very much.
- In the statistical analysis part, the authors should described all analysis methods, such as what kind of statistical test the authors use.
Response: Thank you for the constructive feedback. Accordingly, we have made the following corrections.
“Categorical variables were presented as numbers and percentages, while continuous variables were presented as medians and interquartile ranges (IQR). The chi-squared test or the Fisher’s exact test was used to compare nominal variables. For continuous variables, t-tests or Wilcoxon rank-sum tests were used, as appropriate. Multivariate linear regression analysis and multivariate logistic regression analysis were conducted to examine the factors influencing the development of cognitive biases. The criteria used for each explanatory variable were set as follows: i) factors associated with cognitive biases in previous studies [5–8, 16–19] and ii) univariate screening; factors that were significant at the level of p < 0.05 based on uni-variate regression. Specifically, gender, residents, age, internists, emergency room, night-time, and weekends/holidays were selected in the multiple linear regression analysis for cognitive biases and the multiple logistic regression analysis with and without cognitive bias.”
- The authors should write the variable selection method for the multivariate analysis. The authors described only multivariate analysis results, but the univariate analysis should be also described.
Response: Thank you for your important and constructive input. In response to the feedback, we have added a univariate analysis. We have also changed few numbers, minimum correction, but our main results have not changed. We have corrected all the related descriptions.
Method part
“Categorical variables were presented as numbers and percentages, while continuous variables were presented as medians and interquartile ranges (IQR). The chi-squared test or the Fisher’s exact test was used to compare nominal variables. For continuous variables, t-tests or Wilcoxon rank-sum tests were used, as appropriate. Multivariate linear regression analysis and multivariate logistic regression analysis were conducted to examine the factors influencing the development of cognitive biases. The criteria used for each explanatory variable were set as follows: i) factors associated with cognitive biases in previous studies [5–8, 16–19] and ii) univariate screening; factors that were significant at the level of p < 0.05 based on univariate regression. Specifically, gen-der, residents at the error occurred, internists, emergency room, night-time, and week-ends/holidays were selected in the multiple linear regression analysis for cognitive bias-es and the multiple logistic regression analysis with and without cognitive bias. All p-values < 0.05 were considered statistically significant, and all analyses were conducted using Stata Ver. 14.0 (Stata Corp. 2015. Stata 14 Base Reference Manual).”
Table 6. Contribution to cognitive bias based on univariate and multivariate analysis.
|
Cognitive Biases Contribution (0-10) |
Contributed Cognitive Bias (Yes or No) |
||||||
|
Simple linear regression analysis for cognitive bias contribution level |
Multiple linear regression analysis for cognitive bias contribution level |
Univariate logistic analysis with or without cognitive bias contributing |
Multiple logistic analysis with or without cognitive bias contributing |
||||
|
Coefficient (95% CI) |
p-value |
Coefficient (95% CI) |
p-value |
OR (95% CI) |
p-value |
aOR (95% CI) |
p-value |
Male |
0.29 (-0.75 to 1.34) |
0.579 |
0.56 (-0.49 to 1.61) |
0.290 |
5.63 (1.11 to 28.4) |
0.036 |
1.34 (0.32 to 5.58) |
0.683 |
1-2 years following postgraduation |
-1.0 (-1.94 to -0.50) |
0.039 |
-0.86 (-1.98 to 0.24) |
0.123 |
0.35 (0.12 to 1.02) |
0.054 |
0.12 (0.02 to 0.58) |
0.009 |
Internal medicine |
0.72 (-0.44 to 1.49) |
0.064 |
0.46 (-0.39 to 1.30) |
0.281 |
1.30 (0.48 to 3.57) |
0.602 |
0.75 (0.21 to 2.67) |
0.652 |
Emergency room |
0.62 (-0.27 to 1.51) |
0.169 |
1.04 (0.12 to 1.96) |
0.027 |
2.63 (0.94 to 7.29) |
0.062 |
3.94 (1.15 to 13.51) |
0.029 |
Night shift |
-0.65 (-1.43 to 0.12) |
0.098 |
-0.55 (-1.38 to 0.30) |
0.208 |
1.53 (0.54 to 4.33) |
0.42 |
2.00 (0.56 to 7.32) |
0.280 |
Weekends and holidays |
-0.36 (-1.33 to 0.61) |
0.462 |
-0.34 (-1.33 to 0.65) |
0.500 |
1.32 (0.35 to 4.93) |
0.41 |
1.32 (0.32 to 5.49) |
0.698 |
Table 6
“Significant p-values are presented in bold. Coefficient and adjusted odds ratios (aOR) and 95% confidence intervals (CI) are reported. The variables male, 1−2 years following postgraduation, internal medicine, emergency room, night shift, and weekends and holidays are incorporated in the multiple linear regression analysis for cognitive biases proportion (0-10) and the multiple logistic regression analysis with and without cognitive bias contributing.”
- The authors counted about any bias, but the authors need to analyze the association study if each bias. In addition, the authors should analyze further about the characteristics of physicians who have many errors.
Response: I am very impressed with your very constructive and thoughtful input. We should have indicated our thoughts on what our readers would be interested in. The results are exactly the same as we had analyzed in the last issue, but we have reanalyzed them. As a result, we have incorporated the following text at the end. Regarding the characteristics of physicians, we hope to consider the same in our next research, since the purpose of this study is not to investigate which doctors are most likely to be involved. Thank you for the excellent feedback.
“Finally, we examined each of the 10 cognitive biases used in this study through a multivariate logistic analysis using the same six background factors as above. The results showed statistically significant correlations between errors encountered in the first two years following postgraduation and two cognitive biases, namely, the overconfidence bias (aOR 4.08, 95% CI 1.35 to 12.30, p-value = 0.013) and Hassle bias (aOR 4.45, 95% CI 1.47 to 13.50, p-value = 0.008).
”
- The authors should tabulate about the recommendation/solution to avoid each medical bias.
Response: Thank you for your excellent remark; we had initially attached the Table as a supplemental file. However, other reviewers advised us to include the Table on bias in the main text, thus we have moved it accordingly. We would like to reflect your excellent point in our next paper, which may be a Perspective or Qualitative Research article. Thank you.
#3. (p6) As for anonymity, diagnostic error is sometimes a matter of the institution rather than the individual physician. So how did the authors ensure anonymity of institution in this regard?
Response: Thank you for your inputs. Indeed, it is important to ensure proper anonymity and security for diagnostic error cases.
The survey sheet was submitted to the Ethics Committee of The Medical Research Ethics Committee, Shimane University Faculty of Medicine (No. 20181017-1) before IRB approval was granted. We then started data collection, with the participant filling out the sheet, taking care not to reveal the participant's facility, affiliation, physician's name, or any patient's personal information. The fully anonymized dataset was created by the third author and provided to the first author for analysis. As mentioned above, special care was taken to ensure anonymity. Thank you very much.
#4. (p6) The authors states that data was collected at conferences and meeting places, but the contexts and contents of these meetings should be described more specifically. For example, if this were a voluntary learning community or a special interest group for diagnostic reasoning, selection bias of participants would be inevitable and would undermine the generalizability of the results.
Response: Thank you for these accurate points. We conducted an in-person survey because the narratives include details and painful experiences that we as physicians may not want to generally answer. As you have pointed out, there may be some selection bias for relatively dedicated physicians who are willing to learn more about diagnosis and clinical reasoning in these sessions. Accordingly, detailed information has been added to the Limitation subsection to specify the same.
“The first author held a series of workshops with the same content from July 2018 to January 2019, by which participants could reflect on their diagnostic error cases on the basis of clinical reasoning and learn from the errors. This paper-based data collection process was conducted immediately before the workshop at the place or hospital where the participants worked. Participant recruitment was conducted at conferences in seven different areas of Japan (i.e., Okayama University Hospital in Okayama, Hiroshima City Hospital in Hiroshima, Matsue Seikyo Hospital in Matsue, Shimane University Hospital in Izumo City, an annual meeting of the Japanese Primary Care Association at Osaka, the Japan Institute for Advancement of Medical Education Program in Tokyo, and the Japanese Consortium for General Medicine Teachers in Tokyo), and involved the first author explaining the same methodology each time, and the participants filling out the questionnaire directly.”
P 9, L 285
“The participants who are willing to learn in these clinical reasoning and diagnostic error workshops are selective and likely to be more passionate about the subject than the average physician, and it is questionable whether the results of this study are applicable to other types of physicians.”
#5. (p7) What is the academic significance of asking participants about the number of postgraduate year when the participants experienced the most memorable error? Since they were asked about the time of the "most memorable experience" in their own career, I am wondering what are really academically significant.
Response: Thank you very much. The significance of asking about years since graduation is that the cases that physicians can recall most vividly and in the most detail may have been encountered relatively early in a person's career. Certainly, some physicians may have encountered them the day before or the previous week. In this case, we felt it necessary to evaluate the number of years since graduation because prior studies have indicated that when physicians are relatively inexperienced, errors may be due to knowledge and experience rather than cognitive bias.
#6. (p8) How did the authors confirm that the biases and causal categories the participants responded? If they were wrong, how were they corrected?
Response: Thank you for pointing this out to us. These are some very important questions.
In principle, this method is based on the cases that enable the best recall of the circumstances, environment, and psychological context in which the error occurred. On the other hand, the problem of recall bias is inevitable. Therefore, we have excluded the questionnaires that we judged to be inappropriate.
We have also added this point to the Limitation subsection. Thank you very much.
“The second limitation was concerning recall bias, which implies that participants’ memories may not have been entirely accurate. However, this was an unavoidable limitation due to the self-reflective nature of the survey questionnaire. To extract vivid details, we also required diagnostic error cases in which participants could best recall the circumstances and their psychological states at the time.”
#7. (p8) Please specify the rationale for using p<0.1 as the cutoff for univariate analysis.
Response: Thank you for this comment. We have now rewritten the results of the analysis, including the method of analysis, the conditions for the introduction of variables, and the univariate analysis.
f#8. (p9) Please provide the specific name of the "appropriate" IRB and the approved number (if assigned).
Response: Thank you for your pertinent comment. We have provided the name and the number for the IRB approval of our study in the Methods section and the Declarations as follows.
“The present study was conducted after obtaining approval from The Medical Research Ethics Committee, Shimane University Faculty of Medicine (No. 20181017-1). All participants provided informed consent before participating in the study.”
#9. (p11) The list in Table 4 is about subjective "memorable diagnostic errors," not about the frequency of diagnostic errors among the participant cohort. The authors should describe the results as well as the caption of the table carefully so as not to mislead.
Response: Thank you for your important remarks. We are aware that our English wording could confuse readers. The following is a more detailed explanation of the expression in Japanese, which has been translated. We have made this clear and consistent in all the sentences in the paper. We sincerely appreciate your input.
Revised definition of the most memorable diagnostic error in medicine: "Diagnostic error cases that can be most vividly recalled and described in detail based on the time course, situation, environment, and emotion when the case occurred."
P2 L60
“Participants filled out the survey themselves; the survey was used to collect information on cases defined as the most vividly recalled cases of clear diagnostic errors in a physician’s career, while obtaining detailed situational and environmental information related to the same.”
#10. (p12) The authors almost fail to discuss demographic data and contextual background information. If they are not worth discussing, they should have reflected on the need to collect them for the research question.
Response: Thank you for your feedback. We have rechecked and have made significant changes to several sections of the paper (Results, Tables, Discussion etc.). Presently, these demographic data have been included early in the paper. However, we have clarified in the Conclusions and Discussion sections that the analysis results did not show a statistically significant association with the contribution of cognitive bias to the diagnostic error, especially in this study. We hope that this adequately addresses your comment. Thank you.
Round 2
Reviewer 3 Report
I'm fine with the way the Authors responded my questions.
Reviewer 6 Report
The author revised the manuscript appropriately.